# Third-generation cephalosporin resistant *Escherichia coli* in dogs and cats in Germany in 2019–2021

**Stefanie Katharina Frenzer[1,2], Leonie Feuer[3], Alexander Bartel [1,2], Astrid Bethe[4], Antina Lübke-Becker[2,4], Babette Klein[5], Wolfgang Bäumer[3], Roswitha Merle[1,2] ***

**1** Institute of Veterinary Epidemiology and Biostatistics, School of Veterinary Medicine, Freie Universität Berlin, Berlin, Germany, **2** Veterinary Centre for Resistance Research, Freie Universität Berlin, Berlin, Germany, **3** Institute of Pharmacology and Toxicology, School of Veterinary Medicine, Freie Universität Berlin, Berlin, Germany, **4** Institute of Microbiology and Epizootics, School of Veterinary Medicine, Freie Universität Berlin, Berlin, Germany, **5** LABOKLIN GmbH und Co. KG, Bad Kissingen, Germany

* Roswitha.Merle@fu-berlin.de

**Data Availability Statement:** The data used for the analysis is provided as a csv file in the supplement. The file contains information on species, year, sample type and AST classifications. Geographical

## Abstract

### Objectives

Antimicrobial resistance (AMR) poses a worldwide challenge, threatening global health. The objective of this research was to determine the 3rd generation cephalosporin resistance (3GCR) proportion in *Escherichia (E.) coli* isolated from clinical samples of dogs and cats in Germany.

### Methods

The study utilized result data from antimicrobial susceptibility testing (AST) of isolates obtained from diagnostic samples collected from dogs and cats send in for bacterial examination. Data includes AST results from 3,491 veterinary practices in Germany spanning the years 2019 to 2021, representing 33.1% of practices and clinics nationwide. Out of 175,171 clinical samples, a total of 25,491 *E. coli* strains (14,6%) were evaluated for their susceptibility to antimicrobials, in particular the 3rd generation cephalosporin cefovecin, but also aminoglycosides (gentamicin, GEN), fluoroquinolones (enrofloxacin, ENR), tetracyclines (doxycycline), phenicols (chloramphenicol), folate pathway inhibitors (sulfamethoxazole + trimethoprim), and nitrofurans (nitrofurantoin).

### Results

The cefovecin resistance proportion was 11.6% in the study period. Geographical analysis showed local variations in 3GCR in *E. coli* of ±3%. Regarding all *E. coli* isolates investigated, resistance proportions were observed as follows: 12% for sulfamethoxazole-trimethoprim, 7% for enrofloxacin, 8% for chloramphenicol and 4% for gentamicin. Notably, 3GCR *E. coli* showed significantly higher resistance proportions, specifically 30% for sulfamethoxazole-trimethoprim, 28% for chloramphenicol, 18% for enrofloxacin and 14% for gentamicin.

information is not made available due to privacy reasons.

**Funding:** This work was funded by the Federal Ministry of Food and Agriculture Germany (BMEL) and is part of the HKP-Mon project (FKH: 2820HS002).

**Competing interests:** The authors have declared that no competing interests exist.

## Conclusions

This study represents the first of its kind to utilize an extensive dataset encompassing dogs and cats across Germany. Companion animals have close contact to their owners and transmission of 3GCR between them is likely as well as acquisition from other environmental sources. Resistance proportions (6.7%) against the antibiotic ceftazidime as reported by the German AMR surveillance for human medicine were lower than in our veterinary data. Our study provides an overview of the current 3GCR resistance proportion in Germany and demonstrates the importance of integrated AMR monitoring.

## Introduction

Alarmingly, antibiotic resistance is a leading cause of death worldwide. In 2019, the global impact of antimicrobial resistance (AMR) resulted in estimated 4.95 million deaths. As one of the leading bacterial pathogens, 3rd generation cephalosporin-resistant *Escherichia (E.) coli* (3GCR *E. coli*) played a significant role in causing these fatalities [1]. While antimicrobial resistance is a natural phenomenon, every use of antimicrobial substances including overuse and misuse in human as well as veterinary medicine, contribute to the emergence and spread of antimicrobial resistant bacteria, that may be transmitted directly or indirectly between humans and animals [2]. The transmission of Extended-Spectrum Beta-Lactamase (ESBL)-encoding genes on mobile genetic elements like plasmids allows a dissemination by horizontal gene transfer and thus a rapid spread within the bacterial population [3].

*E. coli* is widely used as indicator species in AMR monitoring systems because 1) especially the commensal *E. coli* lineages are common in the gut of mammals and birds, 2) pathogenic lineages with relevance in human and veterinary medicine do exist [2,4].

In order to combat the worldwide threat of antimicrobial resistant bacterial pathogens, the World Health Organization (WHO), the Food and Agriculture Organization (FAO) of the United Nations and the World Organization for Animal Health (WOAH) launched the Global Action Plan on Antimicrobial Resistance in 2015, which also recommends AMR monitoring in companion animals [5].

Building on this global effort, the European Union (EU) introduced the European One Health Action Plan against Antimicrobial Resistance in 2017 [6]. Within this framework, EU Regulation 2019/6 was implemented in the veterinary sector which mandates the documentation of antibiotic use data, starting for dogs and cats in 2029 [7].

In 2015, Germany submitted the German Antibiotic Resistance Strategy DART 2020 [8] (current version: DART 2030 from 2023) and employs several systems for monitoring AMR, including human clinical isolates "Antibiotika-Resistenz-Surveillance" (ARS) [9] and clinical animal isolates (GERM-Vet) [10].

Due to the close contact between pet animals and their owners, information on resistance-carrying pathogens in companion animals is crucial for human health as well. Knowledge about the occurrence and distribution are needed to gain further understanding and contribute to the One Health aspect of AMR [11].

In this study, we analyzed antimicrobial resistance data of *E. coli* strains isolated from canine and feline clinical samples sent to a large laboratory in Germany from 2019 to 2021. Although some studies exist [12,13], this is the first study comprising a census of diagnostic

data in Germany in this dimension. The objective of the analysis was to determine the 3GCR resistance proportion in clinical *E. coli* isolates from dogs and cats in Germany.

## Material and methods

### Samples and sample processing

Laboklin, an accredited veterinary diagnostic laboratory provided the data on all clinical samples from dogs and cats sent between 2019 and 2021 (175,171 samples). The samples originated from animals presented in veterinary medical facilities (n = 3,491) throughout Germany, representing 33.1% of practices/clinics [14]. Sample types included wound and skin swabs, swabs or lavages from the respiratory and genital tracts, blood and urine specimens. The data set did not contain any information on the time of sample collection or antibiotic pre-treatment. This research was approved by the Central Ethics Committee of Freie Universität Berlin under Approval No. 2021–018.

### Isolation

Bacterial culture was performed using a 3-phase streaking pattern on BD Columbia agar with 5% sheep blood and BD Endo Agar (Becton Dickinson GmbH, Heidelberg, Germany), followed by incubation for 24 h under aerobic conditions at 36˚ C. Enrichment culture was carried out by incubating the swab in Tryptic soy broth (Becton Dickinson GmbH, Heidelberg, Germany) for 24 h at 36˚ C, followed by streaking out on Columbia agar with 5% sheep blood and Endo Agar and incubation under the same conditions. The cultures were classified semi-quantitatively into low, moderate or high growth of bacteria on the primary agar plates and also after enrichment culture. Pure cultures were prepared on a separate blood and Endo Agar plates [15].

### Identification

Identification of bacterial strains was performed by growth morphology, biochemical reactions (oxidase, MAST Diagnostica GmbH, Reinfeld, Germany) and if necessary by mass spectrometry using Matrix-assisted Laser Desorption Ionization Time of Flight Mass Spectrometry (MALDI-TOFMS; Bruker Corporation, Bremen, Germany) [15].

### Antimicrobial susceptibility testing

Antimicrobial susceptibility testing (AST) was performed by the laboratory by determination of the minimal inhibitory concentration (MIC) using Micronaut (MERLIN Gesellschaft für mikrobiologische Diagnostik mbH, Bornheim-Hersel, Germany). A customized panel of antimicrobial substances for gram-negative bacteria was used (see preparation for statistical analysis). Cefovecin is the only 3GC registered for use in cats and dogs in Germany and was therefore the only 3GC included in the panel. The AST results were evaluated in the framework of this study according to clinical breakpoints provided by the Clinical and Laboratory Standards Institute (CLSI) in documents Vet01S ED6 and M100 ED33 and were interpreted into susceptible (S), intermediate (I) and resistant (R) [16,17].

### Preparation for statistical analysis

Based on their relevance for therapy in both small animal and human medicine, the following subset of antimicrobial substances from seven antimicrobial classes were included in our evaluation: beta-lactam antimicrobials (cefovecin, FOV S $\leq$ 2; R $\geq$ 8), aminoglycosides (gentamicin, GEN, S $\leq$ 2; R $\geq$ 8), fluoroquinolones (enrofloxacin, ENR, S $\leq$ 0,5; R $\geq$ 4), tetracyclines

(doxycycline, DOX, S ≤4; R ≥16), phenicols (chloramphenicol, CHL, S ≤8; R ≥32), folate pathway inhibitors (sulfamethoxazole+trimethoprim, SXT, S ≤ 2/38; R ≥ 4/76), and nitrofurans (nitrofurantoin, NIT, S ≤ 32; R ≥ 128).

All samples were assigned to one of six organ system categories defined. Samples associated with wound, musculoskeletal system and surgical samples were grouped under the term "wound". "Skin/soft tissue" (SST) includes skin/soft tissue and secondary reproductive organs, like the mammary gland. "Other" includes other and gastrointestinal tract. Separate categories were "urogenital tract infections" (UTI), "respiratory tract", and "reproductive tract".

### Statistical analyses

All statistical analyses were conducted using R version 4.2.2 (R Foundation Vienna) [18] and the AMR package [19]. Resistance proportions are presented including 95% Wilson confidence intervals (95% CI). The resistance proportion and its CI is calculated for multiple subgroups like by species and by species/sample type. Temporal trends in cefovecin resistance were tested using the Cochrane-Armitage test. To create the density map of the cefovecin resistance, coordinates based on the first 2 digits of the postal codes from the submitting veterinary practice were utilized. The number of resistant samples served as the outcome, and the number of overall samples as the offset for a Poisson regression, to account for the for the number of submitted samples with *E. coli*. The geographical distribution was modelled using a 2-dimensional tensor spline (longitude, latitude). The predicted resistance proportion of the Poisson model was used to visualize the expected local proportion (R package mgcv version 1.8–42 [20]).

### Results

In the three years (2019–2021), a total of 175,171 samples were sent to the laboratory. 27,917 samples (19,154 canine, 8,763 feline) did not yield any growth of specific pathogenic bacterial species.

*E. coli* was identified in 26,429 (20,112 canine, 6,317 feline) samples. Over three years, *E. coli* was isolated in 15.1% of all available samples, with varying proportions between cats (11.7%) and dogs (15.8%). The pathogen was isolated in 14.4% of the 16,111 wound samples, 17.0% of the 21,398 respiratory tract samples, 8.2% of the 67,293 skin/soft tissue samples, 27.1% of the 11,479 UTI samples, 35.2% of 9,428 reproductive tract samples, and 15.3% of the 49,463 other samples. We had 26,180 samples which contained 26,429 isolates. A total of 938 *E. coli* isolates were excluded from further evaluation due to the lack of a valid MIC for cefovecin. The overall number of samples containing one *E. coli* isolate was 25,317 and 87 samples contained two different *E. coli* isolates (*non-haemolytic* & *non-mucoid*, *hemolytic or mucoid*) resulting in 25,491 isolates analyzed (Table 1).

Overall, 11.6% (95% CI 11.2–12.0, n = 2,963) of the 25,491 investigated *E. coli* strains exhibited phenotypic 3GRC resistance. The resistance proportion in dogs of 11.6% (95% CI 11.2–12.1, n = 2,249 of 19,377 isolates) was similar to the 11.7% (95% CI 10.9–12.5, n = 714 of 6,114 isolates) resistance proportion in cats. The 3GCR proportion in our clinical samples was stable between 2019 and 2021 (p = 0.788, Fig 1).

Fig 2 reveals regional differences in 3GCR in *E. coli* of ±3%. Around Bremen and Passau (Bavaria), there were regions with resistance proportions around 15%. Along the former inner-German border (parts of Mecklenburg-Vorpommern, Saxony-Anhalt, Thuringia, parts of Bavaria), resistance proportions were around 10%. In Saarland and the western region around Cologne, Düsseldorf and Duisburg, further regions are noticeable, but with resistance proportions of 12.5%.

**Table 1. Numbers of samples and *E. coli* isolates in dogs and cats collected for the years 2019–2021.**

|  | Overall | Dog | Cat |
|---|---|---|---|
| **Samples** | 175,171 | 122,831 | 52,340 |
| *E. coli* isolates with assessable MIC | 25,491 (100.0) | 19,377 (76.0) | 6,114 (24.0) |
| **Year (%)** |  |  |  |
| • 2019 | 8,952 (35.1) | 6,817 (35.2) | 2,135 (34.9) |
| • 2020 | 8,055 (31.6) | 6,095 (31.4) | 1,960 (32.1) |
| • 2021 | 8,484 (33.3) | 6,465 (33.4) | 2,019 (33.0) |
| **Sample Type (%)** |  |  |  |
| • **Wound** | 2,324 (9.1) | 1,768 (9.1) | 556 (9.1) |
| • **reproductive tract** | 3,319 (13.0) | 3,074 (15.9) | 245 (4.0) |
| • **respiratory tract** | 3,635 (14.3) | 2,211 (11.4) | 1,424 (23.3) |
| • **skin/soft tissue** | 5,517 (21.6) | 4,602 (23.7) | 915 (15.0) |
| • **UTI** | 3,114 (12.2) | 1,898 (9.8) | 1,216 (19.9) |
| • **Other** | 7,582 (29.8) | 5,824 (30.1) | 1,758 (28.7) |

3GCR *E. coli* isolates were more resistant to the non-beta lactam antimicrobials evaluated (Fig 3). Most notable were SXT (12% in total and 30% in 3GCR *E. coli*), DOX (10% vs. 20%) and CHL (12% vs. 30%).

Fig 4 shows that the cefovecin resistance proportions in *E. coli* varied only slightly over the years. The differences within isolates from dogs and cats were marginal. Interestingly, the resistance proportion was lower in isolates originating from the reproductive tract and the urinary tract compared to the other organ systems.

## Discussion

For the first time in Germany, a large-scale analysis was conducted to examine cefovecin resistant *E. coli* in dogs and cats, with cefovecin being the only third-generation cephalosporin (3GC) approved for veterinary use in companion animals. Cefovecin resistance was detected in ESBL and AmpC beta-lactamases producing *E. coli* [22] and Sobkowich et al. were able to

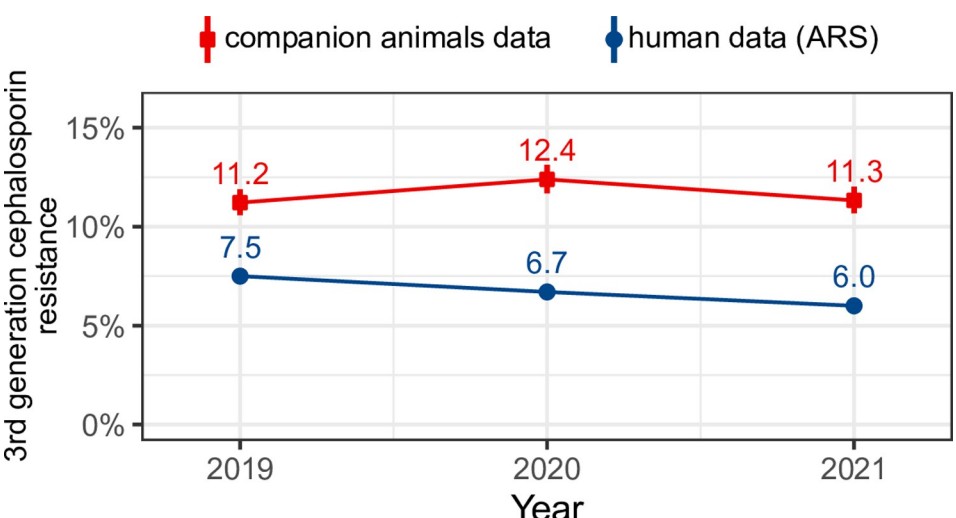

**Fig 1. Percentage and 95% confidence interval of 3GCR isolates of 25,491 *E. coli* from cats and dogs (cefovecin) and from 902,715 isolates from humans (ceftazidime) [9].**

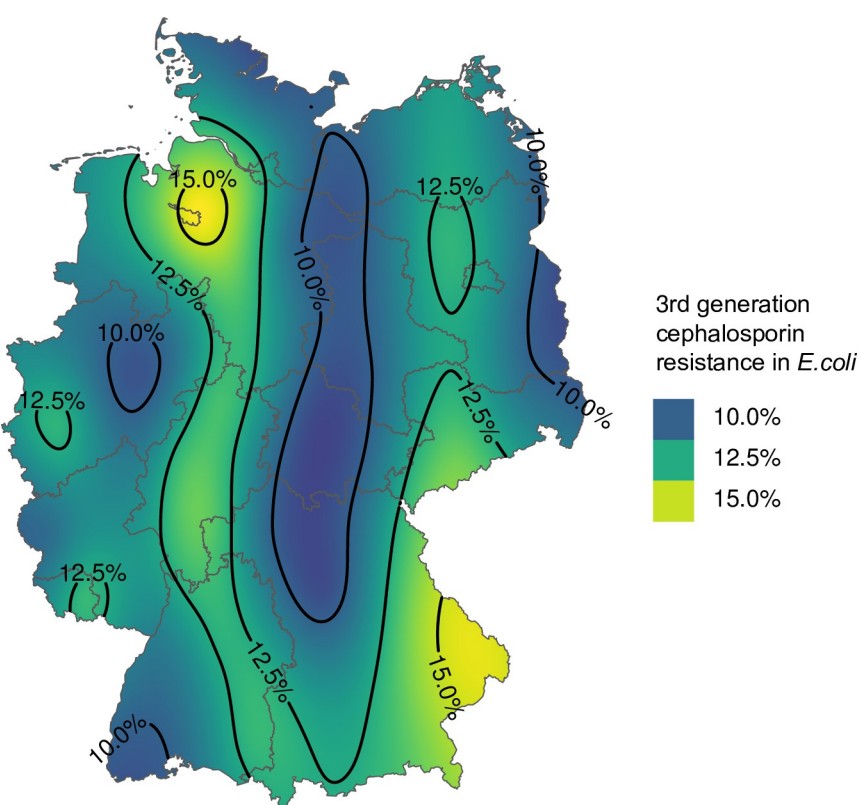

**Fig 2. Distribution of 3rd generation cephalosporin resistant *E. coli* within 25,491 *E. coli* isolates (with assessable MIC) from cats and dogs.** The state borders were provided by GADM for academic use under the gadm license [21].

demonstrate similarity between the results of the phenotypic sensitivity to 3GCs [23]. Therefore, we assume that cefovecin resistance can be reasonably compared with other 3GCs.

Currently, only few studies with a comparably high number of samples exist. Research conducted by Singleton et al. [24] in the UK reported a 3GC resistance proportion of 8.4% in isolates from dogs and 7.2% in isolates from cats. Furthermore, national monitoring systems in countries such as Finland, Denmark, Norway, Sweden, and Switzerland have observed resistance proportions of varying 3GC between 5 and 10% in *E. coli* originating from dogs and cats [25]. In France, surveillance data (n = 7750) from 2014 to 2017 showed a constant 3GC resistance proportion of 6.4% in canine *E. coli*, while in isolates from cats, the resistance proportion declined from 10% to around 4% [26]. The most recent study from North America was also based on routine laboratory data (25 million samples) and reported 17.9% canine and 11.2% feline 3GC resistant isolates [23].

Out of 25,491 *E. coli* isolates in our dataset, 11.6% showed resistance to cefovecin. Consistent with these findings, a study in Australia (n = 855) reported cefovecin resistance proportions of 10.9% in canine *E. coli* and 6.5% in feline *E. coli* [27]. Additionally, recent data from South Korea revealed higher cefovecin resistance proportions by comparison, of 17.1% in *E. coli* isolates (n = 836) from dogs and cats [22].

In terms of comparability, none of the studies except Saputra et al. [27] had information on previous antibiotic treatment, and except for the study by Bourély [26], which only reported resistance proportions in dogs, cats and humans from the urinary tract, the other studies reported various sample types. Most studies reported resistance proportions following CLSI guidelines [22,23]. Singleton et al. [24] provided data from numerous veterinary diagnostic

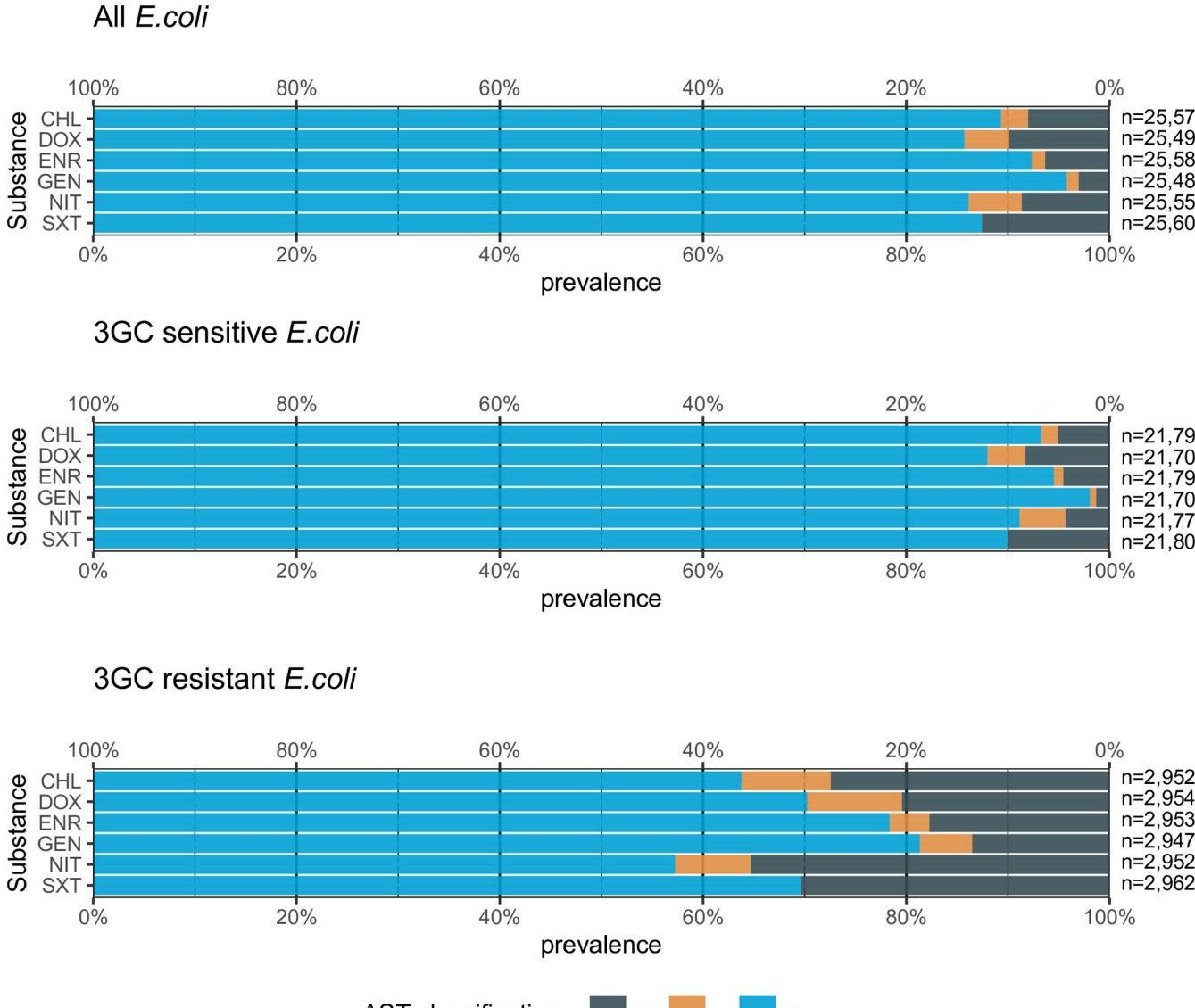

**Fig 3. Antimicrobial susceptibility patterns of the substances chloramphenicol (CHL), doxycycline (DOX), enrofloxacin (ENR), gentamicin (GEN), nitrofurantoin (NIT) and sulfamethoxazole/trimethoprim (SXT) in all 25,491 *E. coli* isolates and in 2,963 3GCR *E. coli* isolates from 175,171 samples from cats and dogs.** Abbreviations: S–sensitive, I–intermediate, R–Resistant.

laboratories across the UK and reported resistance proportions according to CLSI, EUCAST, and BSAC guidelines. Similarly, Bourély et al., in their nationwide study across France, reported resistance proportions based on EUCAST and CA-SFM guidelines. Saputra et al. used CLSI guidelines to report their resistance proportions but also included epidemiological cut-off values (ECOFFs). None of the studies cited reported data from selective isolation. Instead, they obtained routine results from veterinary diagnostic laboratories without pre-selection. All these points must be considered regarding comparability.

In Germany, GERM-Vet serves as the only national monitoring system in the companion animal sector [28]. GERM-Vet uses an active monitoring approach by collecting isolates from commercial veterinary laboratories in Germany. This active approach provides genotypical analysis for ESBL-producing bacteria. In 2021, *E. coli* isolated from gastrointestinal and

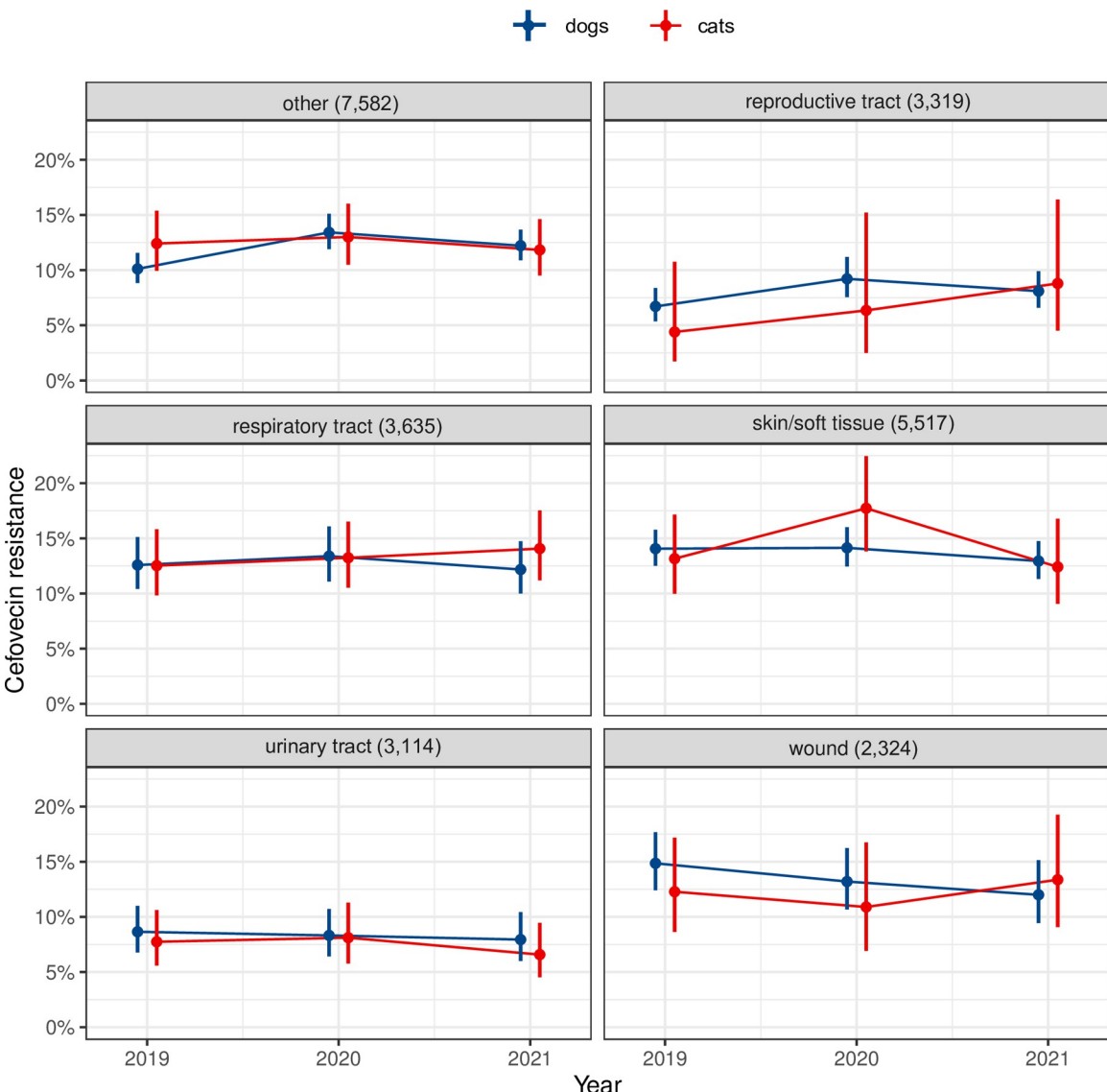

**Fig 4. Percentage of 3GC resistant isolates per organ system in 25,491 *E. coli* isolates from dogs and cats.**

urinary tract from diseased dogs and cats were examined. Only 2,7% of dog isolates (n = 182) and 4.0% of cat isolates (n = 99) were ESBL-producers. However, this active approach results in a smaller number of isolates in GERM-Vet, compared to our passive monitoring approach. This allows for a more detailed characterization of isolates, but also increases sampling uncertainty and limits representativeness, if the active sampling is not random.

The ARS data showed an overall resistance proportion of 6.7% for ceftazidime in humans in 2020, declining over time [9]. Ceftazidime is used in human medicine for severe infections such as urinary tract infections, intra-abdominal infections, or pneumonia, especially in intensive care [29]. In veterinary medicine, cefovecin is a long-acting drug (administered for 14 days) that is often used in cats for fight injuries, abscesses or urinal tract infections. It has to be applied only once by the veterinarian, prevents application mistakes by the owners and reduces stress for pets and owners [30,31]. However, some factors may affect the accuracy of our results. One potential issue is an upward sampling bias, as veterinarians may only test for

susceptibility when there is reasonable suspicion of resistance, due to cost considerations [23]. Additionally, duplicated isolates in the study data, which could not be removed, may also influence the results.

The origin of antimicrobial resistant bacteria is multifaceted [32]. Studies show, that 6.3–10.3% of adults and 2.3% of children in Germany are asymptomatic ESBL- *E. coli* carriers [33]. Risk factors are antimicrobial use, person-to-person contact, international traveling, contaminated food and contact to livestock or companion animals [32,33]. For dogs and cats its similar, but some of them have a more direct contact to the environment, including contact to other animals' excretions [34]. Another possible source for the transmission of AMR are raw meat based diets, that do not meet hygiene standards [35]. Furthermore, the close contact between companion animals and their owners can facilitate the exchange of bacteria, including resistant strains, making it a One Health issue. Considering the similar living conditions and close contact between cats/dogs and their owners (shared living space, sharing of furniture and possibly the bed), the possibility of transmission of resistant bacteria is given. However, further research is needed to assess the actual extent of the exchange of 3GC resistance [11].

Analyzing the geographical distribution of 3GC resistance in dogs and cats across Germany, notable areas of elevated resistance proportion were found near Bremen and Passau (Bavaria), with resistance proportions around 15%. Whether these geographical differences are stable over time remains to be seen, since only then, a connection to geographical characteristics can be considered. Comparison to human medicine was not conducted since high-resolution geographical information was not available in the ARS-database [9]. Given the population and livestock density in Germany, it is possible to hypothesize whether there is a link [36,37].

Our study revealed that 3GCR *E. coli* isolates had more co-resistances to other substances (Fig 3), which is not surprising as ESBLs are often encoded on large plasmids facilitating horizontal transmission and carry genes for further resistance properties [2].

The lower resistance proportions in the reproductive and urogenital system are consistent with other research [38]. In cases of UTI where the available options of first-line antibiotics have been exhausted, 3GC, fluoroquinolones or nitrofurantoin remain alternatives. Since cefovecin needs to be applied only once, animal owners might be more motivated to pay for the mandatory AST. Thus, the coverage of tests among UTI is higher reducing the sampling bias.

Monitoring antibiotic resistance plays a pivotal role in antibiotic stewardship, as it enables to assess the impact of measures regarding AMU on the development of resistance proportions [6]. In human medicine, Germany has established prescription guidelines for empirical antibiotic therapy [39], which helps practitioners in decision making regarding their antibiotic therapy and provides legal certainty. The development and introduction of comparable guidelines also in veterinary medicine could make a significant contribution to the rational use of antibiotics. Therefore, detailed and current data in AMR rates is urgently needed.

An active monitoring approach has been successfully established in Germany with GERM-Vet, which can monitor for changes in genotypes underlying 3GCR. Active monitoring can be associated with substantial cost and is therefore often limited to smaller sample sizes. Passive monitoring is cost-effective as existing AST results are used and no additional laboratory analyses are required. Our study shows a complementary passive monitoring approach, which provides a survey of one third of German veterinary practices and thus provides more representative results and enables a more detailed temporal and spatial analysis [40–42]. Results from the passive monitoring could also be used to plan the active sampling strategy, as, at least in Germany, both isolates and AST results are collected based on clinical samples.

## Conclusions

Accurate knowledge of antibiotic resistance in animal pathogens is crucial for the optimal use of antibiotics and benefits human and animal health. Our study could show that it makes sense to include a passive monitoring based on accredited veterinary laboratories on a voluntary basis–possibly with incentives to extend the panel by ceftazidime and cefotaxime as ESBL confirmation tests–into active national monitoring systems (GERM-Vet in Germany). Although these data may have restrictions concerning meta-data, in our opinion, this disadvantage is outweighed by the amount and coverage of data and the cost-effectiveness.

Ideally, interdisciplinary collaboration is important to achieve integrated AMR surveillance combining data from humans, animals, and the environment. Therefore, we support the approaches to establish an EU-wide AMR surveillance for the veterinary sector.

## Supporting information

**S1 File. Raw data.**
(CSV)

## Acknowledgments

Our thanks go to the laboratory company Laboklin GmbH & Co.KG, Bad Kissingen, namely Dr. Marianne Schneider, for providing these data and assisting in data analysis.

## Author Contributions

**Conceptualization:** Stefanie Katharina Frenzer, Leonie Feuer, Alexander Bartel, Roswitha Merle.

**Data curation:** Stefanie Katharina Frenzer, Leonie Feuer, Alexander Bartel, Astrid Bethe, Babette Klein.

**Formal analysis:** Stefanie Katharina Frenzer, Leonie Feuer, Alexander Bartel.

**Investigation:** Babette Klein.

**Methodology:** Stefanie Katharina Frenzer, Leonie Feuer, Alexander Bartel.

**Project administration:** Roswitha Merle.

**Software:** Alexander Bartel.

**Supervision:** Wolfgang Bäumer, Roswitha Merle.

**Validation:** Antina Lübke-Becker, Babette Klein.

**Visualization:** Alexander Bartel.

**Writing – original draft:** Stefanie Katharina Frenzer.

**Writing – review & editing:** Alexander Bartel, Astrid Bethe, Antina Lübke-Becker, Babette Klein, Wolfgang Bäumer, Roswitha Merle.

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
