## [Decision Letter · Decision Letter 0]

14 Feb 2024

PONE-D-23-35833Towards integrated AMR monitoring: Third-generation cephalosporin resistant Escherichia coli in dogs and cats in Germany in 2019-2021PLOS ONE

Dear Dr. Bartel,

Thank you for submitting your manuscript to PLOS ONE. After careful consideration, we feel that it has merit but does not fully meet PLOS ONE’s publication criteria as it currently stands. Therefore, we invite you to submit a revised version of the manuscript that addresses the points raised during the review process.

As you can see there are quite a number of comments and doubts about the manuscript. Please reply to them all. As an addition, please mention the differences between a passive surveillance as opposed to an active surveillance. Mention how you can compare the data of those and what are the pitfalls. It will help clarifying the dataset you used. Also include more information on cefovecin and the action of ESBLs on this cephalosporin, some references would be good.

We look forward to receiving your revised manuscript.

Kind regards,

Patrick Butaye, DVM, PhD

Academic Editor

PLOS ONE

 [This work was funded by the Federal Ministry of Food and Agriculture Germany (BMEL) and is part of 

the HKP-Mon project (FKH: 2820HS002)].  

4. Please include the reference section of your manuscript.

5. We note that Figure 2 in your submission contain copyrighted images. All PLOS content is published under the Creative Commons Attribution License (CC BY 4.0), which means that the manuscript, images, and Supporting Information files will be freely available online, and any third party is permitted to access, download, copy, distribute, and use these materials in any way, even commercially, with proper attribution. For more information, see our copyright guidelines: http://journals.plos.org/plosone/s/licenses-and-copyright.

Reviewers' comments:

Reviewer's Responses to Questions

**Comments to the Author**

1. Is the manuscript technically sound, and do the data support the conclusions?

Reviewer #1: Partly

Reviewer #2: Yes

Reviewer #3: Partly

2. Has the statistical analysis been performed appropriately and rigorously? 

Reviewer #1: No

Reviewer #2: No

Reviewer #3: Yes

3. Have the authors made all data underlying the findings in their manuscript fully available?

Reviewer #1: Yes

Reviewer #2: Yes

Reviewer #3: No

4. Is the manuscript presented in an intelligible fashion and written in standard English?

Reviewer #1: Yes

Reviewer #2: Yes

Reviewer #3: Yes

5. Review Comments to the Author

Reviewer #1: The title can be misleading as readers would expect some real policy/action research done, along with this descriptive study – I suggest removing “Towards integrated AMR monitoring:” – this study is a simple description of collected data over three years.

I suggest using ‘resistance proportion’ instead of ‘resistance rate’ throughout the text.

Line 33: the abstract needs a complete re-write.

For instance, I do not see any real “objective” stated in lines 34-37; I cannot get what you have done in Lines 38-41 – wasn’t the lab work already done and you used the electronic data? This is not the methodology for your research …

Line 86-91: please clearly state your objective – which, from my perspective, is to describe the resistance to 3GC (and others?) in clinical E. coli isolates from dogs and cats in Germany, 2019-2021… or something similar.

Line 136: In general, the statistical analysis section is vague and incomplete – it should be aligned with the results – in the current form, I cannot follow

Line 147: the authors mention building a “Poisson Regression” model (in Line 141), but I do not see any results (/table) associated with this model?

Line 154: starting sentence with a number (776)

Line 157: complete ‘Table 1’ in a scientific manner and based on the journal’s guidelines; e.g., clearly define what numbers in parentheses are, or ‘#’, what are 1 and 2 under “E. coli with MIC per Sample (%)”?

Line 159: Why should you even compare human and animal %s directly with each other? Directly comparing human and any animal spp. %s can be misleading – the sources and contamination/infection, strains, and epidemiology of AMR in these species are completely different – the only useful piece of information from this comparison is looking at correlations over long time periods – which still can be misleading. The discussion section should also shift the focus away from this (see below). I suggest removing this figure and provide a short discussion around the correlations between human and animal figures instead (if similar methodologies were used)

Line 166: Please remove Figure 2; we do not get any useful information from Figure 2. You have the description in the text – it just looks beautiful. Also, I’m not even sure 10% to 15% difference is worth emphasizing much or discussing anywhere.

Lines 202-207: this paragraph is not relevant to the work done in this study – while it is a good and true statement in general

Line 214: I agree with “Direct comparisons to the results in our study cannot be done because not all 3GCR E. coli are ESBLs, and we reported 3GCR” – so why do you discuss (compare) this?

Line 222: I do not follow: “but “data […] are too scarce to come to a definite answer””! was this supposed to be a complete sentence?

Line 223-229: as mentioned above, I do not know why even we should compare human %s with animal % directly. Please either remove this completely or re-word so it does not sound like we expect the same rate/proportion of resistance in different species

Line 232: as it should not be done “Comparison to human medicine was not conducted due to different data resolution”

Line 246: please be very clear and briefly explain what you mean by “sampling bias” and do not repeatedly refer to other sections in the discussion (remove: ‘above-mentioned …’ or ‘see above’).

Line 254-259: not sure how this relates to the work you have done in this study

Line 261: the conclusions are irrelevant to the research conducted – please at least try to connect your own findings (3GCR E. coli) with the practical implications – not just stating some general facts about antimicrobial stewardship programs and what we need to do.

Reviewer #2: I enjoyed reading your article. Please find below my comments.

1) I think the title and the abstract does not reflect fully the article content. The article describes all the resistance patterns to several antimicrobial class and does not solely focuses on 3rd generation cephalosporin resistance in E. coli isolates.

2) Line 127 "seven drug families" should be changed to "seven antimicrobial classes"

3) Could you clarify why the systemic were grouped under the term “wound”? What systemic means? Isolation from blood? Systemic might be classified under other.

4) Would you expect that your E.coli isolates were pathogenic? Did the lab tested for virulence? I would expect that the site of isolation would point to the pathogenicity of the isolate. An isolate identified in organs or blood would have a higher probability to be pathogenic than one that was isolated from skin. On this note, do you know if the sample was obtained from free-catch urine or via cystocentesis for the E.coli isolated from UTI cases. Free catch would have more commensal E. coli.

5) Line 164. When you compare rates between years you should use a trend test. You have two options: Mann-Kendall Test For Monotonic Trend or Cochran–Armitage test for trend.

6) Line 138 - “create the density map, coordinates based on the first 2 digits of the postal codes from the submitting veterinary practice were utilized.” Please define what metod was used? Did you account for the background cat and dog populations? You would expect more samples and higher resistance rates from areas with high number of dogs. In the US research studies estimated the number of dogs and cats based on the size of households. I think that would be possible to do it in Germany too.

7) I suggest that Fig 3 and Fig 4 –should be redone and make new figures by presenting the cat and dog data separately.

8) As Plos One is an international journal I suggest to expand the discussion section by comparing your results to other studies from North America, Australia, etc.

Reviewer #3: This study is built on a comprehensive data set from a clinical laboratory in regard of the phenotypic resistance of E. coli isolates, especially in regard of resistance to cefovecin. Surveillance data on AMR in companion animals are rare and capacities should be extended.

The major issue of this study is the comparability to other datasets. To assess the resistance to 3rd generation cephalosporins (3GCR), which is mainly caused by ESBL or AmpC beta-lactamases, the antibiotic cefovecin was used. This might be due to the recommendations of the German Veterinary Society, although the authors didn’t explicit mentioned. Nevertheless, the study misses to show that cefovecin resistance is comparable to typical 3GC cefotaxime or ceftazidime or that cefovecin resistance is mediated by ESBL/AmpC beta-lactamases. At least the authors could have shown, that ESBL/AmpC-producing E. coli show phenotypic resistance to cefovecin. As there are authors from the Institute of Microbiology as well as from the Veterinary Centre for Resistance Research, they have the opportunities for this (proven by other publications). It is essential to investigate these points, as cefovecin isn’t used in other German surveillance systems (e.g. ARS in humans;) or monitoring systems (Zoonoses Monitoring; healthy animals and food; GERM-Vet in diseased animals). Otherwise, data is not comparable to the different systems and therefore not suitable as a OneHealth approach. No EUCAST definitions for E. coli and cefovecin are available. EUCAST definitions are used in EU surveillance/monitoring programs, which makes it hard to compare the data.

The discussion is inconsistent at this point. Overall majority of publications on 3rd generation cephalosporin resistance (3GCR) are on ESBL/AmpC producing bacteria. They discuss cefovecin resistance rates to those of ESBL/AmpC rates (based on cefotaxime and ceftazidime resistance, although they also discuss, that there might be differences between these two terminologies. Further, they say in line 251, that FOV resistance not necessarily mean resistance to ceftazidime, which indicates also not necessarily to cefotaxime both indicators for 3. generation cephalosporin resistance.

Further as the different systems use different values (clinical breakpoint vs. ECOFF) the original MIC values are needed to give in Suppl. Table and not only interpretation. Otherwise, it is not possible to compare resistance rates (other than cefovecin) between different sectors

Another inconsistency is the usage of the terms “monitoring” and “surveillance”. Even between the headline (monitoring) and the abstract (surveillance) they change the wording. It is strongly recommended to change the headline as this was not a monitoring (or towards a monitoring) and use the terms appropriate throughout the manuscript. The authors also could state out more clearly, why there is a need for a surveillance system, which can’t be used for ESBL/AmpC comparison between the sectors (without showing that it is comparable) while there is a monitoring on diseased animals since more than 15 years. Wouldn’t it be more efficient to extend this monitoring when there is a need of more data points?

There is a lack of discussion, why the resistance rates are higher than in other studies. Might this be not only because of ESBL/cefovecin issue but also on the data set? It seems that all available data was used, but it wasn’t evaluated whether they received an antibiotic treatment before.

Manuscript:

Line 52 (and other): this is not a prevalence of 3GCR in cats and dogs; there is the bias of diseased animals. For an overall prevalence, the basic population should be just cats and dogs. Better would be “resistance rate”.

Line 90/91: it is not in the sense of a OneHealth concept, when the data is not comparable between the sectors

Line 94: in this section you should mention, if there is anything known of sampling time point. Were the samples taken prior to a treatment? Was it possible to exclude animals that received antibiotics in a defined period before these samples were taken? This should also be discussed.

Line120: The reference 14 is not appropriate; the panel is not given there, just the resistances found. Please provide here a table with the panel and the used breakpoint values. Is this the panel recommended by DVG? This can be combined with line 128-130

Line 125 / Table1: Please provide information, if there was an exclusion of duplications. Did you exclude isolates, which were taken from the same animals at several time points or from different organs at the same time (line 228 says it was not excluded?)? Is this indicated in Table 1 with “E.coli with MIC per sample”? This part of the table is not clear and also not explained in the text or the table description. In the method part of the abstract, you mentioned 26,180 isolates, but you have “only” 25,404 isolates with an assessable MIC and if you exclude duplications 25,317. Which number did you use to calculate resistance rates?

Line 160: Figure description is not correct; y-axes is only on cefovecin; here again it is inconsistent that you compare cefovecin resistance and ceftazidime resistance but in line 251 you say that there might be differences. As long as you didn’t show, that cefovecin and ceftazidime resistance can be compared, you shouldn’t provide a figure on this, as this indicated a not proven comparability to the reader.

Line 171: when the average resistance rate is 11.6% than 10% resistance rate is not low; it is only “lower”. This is a nice figure, but only poor discussion on the findings. Is it possible to map this to other factors like land use; livestock density or other?

Line 196: why Enterobacterales? The reference (21) provides data on E. coli.

This part of the discussion: you switch between FOV resistance and 3GCR; please provide species and investigated antimicrobial substance for the studies you mention (e.g., line 201)

Line 204: [25] Reference information given not enough, hard to find; please provide a link https://iris.who.int/handle/10665/340079. ; Tricycle is on ESBL E.coli and conformation should include cefotaxime or ceftriaxone and ceftazidime resistance testing; which is not in concordance to your study.

Line 209: reference 10 is not the latest report, please you the latest data for the discussion

Line 213-214: inconsistent; on the one hand you compare your data to other studies in other parts of the discussion but here you say it is not directly comparable; see above; check the comparability with a subset of isolates (and with a range of different ESBL at least CTX-M, SHV, TEM, CMY)

Line 225-226: that is to general; urban cats and dogs contact to other envrionent than animals in the country side; the influence of human and livestock would be different; could you see realtions to one of the factors? Figure 1 doesn’t implicate this (see below).

Line 230-232: you just metion your findings but there is no discussion on that

Line 239-241: in contrast to line 250 , where you say you don’t know ESBL status (although ESBL might be the most reasonable resistance mechanism)

Discussion on a one health context: you only referre to humans and cats and dogs there is no discussion on German livestock (healthy animals as well as diseased animals)

Conclusion:

Be careful with monitoring and surveillance; there is a monitoring system for diseased companion animals;

This is not a discussion; avoid references here; you also didn’t discuss on livestock resistance before and further, in Germany resistance rate to cefotaxime in commensal E. coli are lower than the cefovecin resistance rate in your study on cats and dogs and the references should be on the German monitoring systems for appropriate comparison.

6. PLOS authors have the option to publish the peer review history of their article (what does this mean?). If published, this will include your full peer review and any attached files.

Reviewer #1: No

Reviewer #2: No

Reviewer #3: No

---

## [Author Response · Author response to Decision Letter 0]

10 Apr 2024

Response to the reviewers 

Editor

As you can see there are quite a number of comments and doubts about the manuscript. Please reply to them all. As an addition, please mention the differences between a passive surveillance as opposed to an active surveillance. Mention how you can compare the data of those and what are the pitfalls. It will help clarifying the dataset you used. 

AU: We added the differentiation between active and passive surveillance/monitoring to the discussion. A separate paragraph was added, discussing the potential for combining an active and passive surveillance, without taking resources away from the active surveillance.

Also include more information on cefovecin and the action of ESBLs on this cephalosporin, some references would be good.

AU: Cefovecin is a long-acting third-generation cephalosporin (trade name Convenia). We added some studies which show that the mechanisms for Cefovecin resistance are the same as for other third-generation cephalosporins, namely primarily ESBL and to a lower extent AmpC beta-lactamases. The predominant resistance gene group seems to be blaCTX-M [1,2].

Reviewer #1: 

Thank you for taking time for reviewing our paper and your valuable feedback.

The title can be misleading as readers would expect some real policy/action research done, along with this descriptive study – I suggest removing “Towards integrated AMR monitoring:” – this study is a simple description of collected data over three years.

AU: The title has been changed. (line 3)

I suggest using ‘resistance proportion’ instead of ‘resistance rate’ throughout the text.

AU: Thank you for your advice! Reviewer 3 suggested to use resistance rate instead of prevalence, so this is what we did. 

Line 33: the abstract needs a complete re-write.

For instance, I do not see any real “objective” stated in lines 34-37; I cannot get what you have done in Lines 38-41 – wasn’t the lab work already done and you used the electronic data? This is not the methodology for your research …

AU: The objective and methods were changed. Yes, we only used the electronic AST result data. We agree it was misleading before. (lines 34-44)

Line 86-91: please clearly state your objective – which, from my perspective, is to describe the resistance to 3GC (and others?) in clinical E. coli isolates from dogs and cats in Germany, 2019-2021… or something similar.

AU: Thank you for this comment, done. (lines 90,91)

Line 136: In general, the statistical analysis section is vague and incomplete – it should be aligned with the results – in the current form, I cannot follow.

AU: The statistical analysis is mostly limited to the calculation of confidence intervals (CI) for the resistance proportion. The resistance proportion and its CI is calculated for multiple subgroups like by species and by species/sample type. The second part of the statistical analyses relates to the geographical analyses (see next answer). (lines 140-149)

Line 147: the authors mention building a “Poisson Regression” model (in line 141), but I do not see any results (/table) associated with this model?

AU: The poisson model was used to fit a 2 dimensional spline (latitude and longitude). Splines have a large number of parameters, which cannot be presented as a table. The spline was only fitted to estimate a generalized and smoothed resistance proportion for any point in Germany. The poisson model allows to weight the calculated proportions according to the number of available AST results by using an offset. Figure 2 presents the results of the poisson spline model graphically. (lines 145-149)

Line 154: starting sentence with a number (776)

AU: Thank you, it has been changed. (lines 157-158)

Line 157: complete ‘Table 1’ in a scientific manner and based on the journal’s guidelines; e.g., clearly define what numbers in parentheses are, or ‘#’, what are 1 and 2 under “E. coli with MIC per Sample (%)”?

AU: Regarding the line labeled "#E.coli with MIC per Sample (%)," this indicates that each sample in line "1" contained one E.coli isolate (n = 25,317). In row "2", 87 samples contained an additional isolate alongside E. coli, either “E.coli mucoid” or “E.coli hemolytic”. We added an explanation in the description of the table. (lines 162-163)

Line 159: Why should you even compare human and animal %s directly with each other? Directly comparing human and any animal spp. %s can be misleading – the sources and contamination/infection, strains, and epidemiology of AMR in these species are completely different – the only useful piece of information from this comparison is looking at correlations over long time periods – which still can be misleading. The discussion section should also shift the focus away from this (see below). I suggest removing this figure and provide a short discussion around the correlations between human and animal figures instead (if similar methodologies were used)

AU: Thank you for your comment! We are aware of this challenge. You are of course right that humans and animals are different species. However, today we live in such close contact with our pets that we assume that transmission of E. coli occurs between pets and their owners, which means there are shared sources of contamination and infection. Therefore, we believe the comparison is useful under the One Health framework. Of course, there are differences in sampling bias and antibiotic usage, which we discussed. In addition, the AST results in veterinary and human medicine both influence the use of a reserve antibiotic. The resistance data for human medicine (ARS database) were also collected from routine laboratory testing under same passive surveillance sampling methodology as our data. (lines 235-247)

Line 166: Please remove Figure 2; we do not get any useful information from Figure 2. You have the description in the text – it just looks beautiful. Also, I’m not even sure 10% to 15% difference is worth emphasizing much or discussing anywhere.

AU: A 10 to 15% difference corresponds to a relative risk/ prevalence ratio of 1.5, which is commonly assumed to be a medium effect size. Nevertheless, even if no geographical variation could be observed, we think null effects should still be reported. (line 174)

Lines 202-207: this paragraph is not relevant to the work done in this study – while it is a good and true statement in general.

AU: The sentence has been removed.

Line 214: I agree with “Direct comparisons to the results in our study cannot be done because not all 3GCR E. coli are ESBLs, and we reported 3GCR” – so why do you discuss (compare) this?

AU: Thank you for this comment, we clarified the statement. Since the same resistance mechanisms seem to present in different 3GCs, there is a comparability. (lines 204-208)

Line 222: I do not follow: “but “data […] are too scarce to come to a definite answer””! was this supposed to be a complete sentence?

AU: You are right, this sentence was a bit confusing. It has been changed. (lines 233-234)

Line 223-229: as mentioned above, I do not know why even we should compare human %s with animal % directly. Please either remove this completely or re-word so it does not sound like we expect the same rate/proportion of resistance in different species

AU: In the One Health framework human, veterinary and environmental aspects of transmission of AMR should be considered. Close contact between companion animals and owners is given. Despite difference in pathomechanisms a comparison under these circumstances still seems useful to us. (lines 235-241)

Line 232: as it should not be done “Comparison to human medicine was not conducted due to different data resolution”

AU: We have refrained from making a comparison here because we only had a low-resolution geographical division of the ARS data into north-east, north-west, south-east, south-west and west. (lines 251-253)

Line 246: please be very clear and briefly explain what you mean by “sampling bias” and do not repeatedly refer to other sections in the discussion (remove: ‘above-mentioned …’ or ‘see above’).

AU: Sampling bias in our case means that there is a possibility that there is a bias because vets often only do AST if the first antibiotic does not work or the animal's condition worsens and not before each antibiotic treatment. Therefore, the proportion of 3GCR E. coli may appear higher than it actually is. 

“above-mentioned and see above” have been deleted. (lines 260-261)

Line 254-259: not sure how this relates to the work you have done in this study

AU: As discussed AMR monitoring is one aspect of antibiotic stewardship. We added a clarification sentence at the end of this paragraph. (lines 267-268)

Line 261: the conclusions are irrelevant to the research conducted – please at least try to connect your own findings (3GCR E. coli) with the practical implications – not just stating some general facts about antimicrobial stewardship programs and what we need to do.

AU: We reformulated our conclusions part to link them to our study findings. We hope that it is to your satisfaction. (lines 281-290)

Reviewer #2: I enjoyed reading your article. Please find below my comments.

Thank you very much for taking the time to read our article and for your valuable suggestions.

1) I think the title and the abstract does not reflect fully the article content. The article describes all the resistance patterns to several antimicrobial class and does not solely focuses on 3rd generation cephalosporin resistance in E. coli isolates.

AU: Many thanks for the advice. We have revised the abstract. (lines 34-56)

2) Line 127 "seven drug families" should be changed to "seven antimicrobial classes"

AU: Thank you! It has been changed. (line 128)

3) Could you clarify why the systemic were grouped under the term “wound”? What systemic means? Isolation from blood? Systemic might be classified under other.

AU: Thank you for the comment. That's right, it's a bit misleading. Systemic includes samples such as ascites, chest/abdominal cavity punctate or thoracic effusion. We have therefore renamed it to surgical samples. (line 135)

4) Would you expect that your E.coli isolates were pathogenic? Did the lab tested for virulence? I would expect that the site of isolation would point to the pathogenicity of the isolate. An isolate identified in organs or blood would have a higher probability to be pathogenic than one that was isolated from skin. On this note, do you know if the sample was obtained from free-catch urine or via cystocentesis for the E.coli isolated from UTI cases. Free catch would have more commensal E. coli.

AU: Dear reviewer, thank you for this valuable comment. No, we would not necessarily expect the isolated E. coli to be pathogenic. We have not received any tests for virulence. Some of the samples we received were cystocentesis urine, but some were just urine. This was summarized under "UTI". We agree that free catch urine would contain more commensal E. coli, but this distinction was not made here. We would also not assume that E. coli from collected urine is always a contamination. Mixed culture: yes/no, how many bacterial species and the amount of bacteria also play an important role in the evaluation. (line 191)

5) Line 164. When you compare rates between years you should use a trend test. You have two options: Mann-Kendall Test For Monotonic Trend or Cochran–Armitage test for trend.

AU: We added the results for the Cochrane-Armitage test (p=0.788). (line 173)

6) Line 138 - “create the density map, coordinates based on the first 2 digits of the postal codes from the submit-ting veterinary practice were utilized.” Please define what method was used? Did you account for the background cat and dog populations? You would expect more samples and higher resistance rates from areas with high num-ber of dogs. In the US research studies estimated the number of dogs and cats based on the size of households. I think that would be possible to do it in Germany too.

AU: We used a 2-dimensional spline (latitude, longitude) and fitted it using a poisson regression. The poisson regression accounts for the number of submitted samples with E. coli (offset). We amended the statistical method. Since we only calculated the resistance rate and did no risk factor analysis we only accounted for the number of submitted samples (i.e. number of infections with E. coli). But since the data was collected on postal code level, it is indirectly adjusted for the population density and we could assume that pet density and population density are highly correlated. Postal codes are smaller for areas with high population density and larger for low population density areas (the areas are designed to have approximately the same mail volume). A more sophisticated ad-justment was not done for Germany yet and reliable data is not available. We would therefore prefer to keep the analysis as it is. (lines 140-149, line 174)

7) I suggest that Fig 3 and Fig 4 –should be redone and make new figures by presenting the cat and dog data separately.

AU: Yes, you are right. Figure 4 is displayed separately for dogs and cats in figure S1 (at the moment supplementary figure). Would you prefer supplementary figure S1 instead of figure 4? (line 191)

8) As Plos One is an international journal I suggest to expand the discussion section by comparing your results to other studies from North America, Australia, etc.

AU: We have expanded the discussion with some international studies. (lines 202, 205, 207)

Reviewer #3: This study is built on a comprehensive data set from a clinical laboratory in regard of the phenotypic resistance of E. coli isolates, especially in regard of resistance to cefovecin. Surveillance data on AMR in companion animals are rare and capacities should be extended.

The major issue of this study is the comparability to other datasets. To assess the resistance to 3rd generation cephalosporins (3GCR), which is mainly caused by ESBL or AmpC beta-lactamases, the antibiotic cefovecin was used. This might be due to the recommendations of the German Veterinary Society, although the authors didn’t explicit mentioned. 

AU: Thank you for your valuable feedback. You are correct in pointing out the contradictions present in the initial version. We think that cefovecin resistance is comparable to resistance to other cephalosporins of the third generation. The most common mechanism is certainly beta-lactamase production (ESBLs) or the production of acquired AmpC cephalosporinases. It's important to note that the data utilized in this study were not specifically gathered for this research but were sourced from a comprehensive dataset obtained through routine diagnostic testing. As cefovecin is the only 3GC in our dataset, that is approved for dogs and cats in Germany, it is tested here. While it's true that cefovecin is not included in ESBL diagnostics, existing studies suggest the presence of similar resistance mechanisms [1]. To answer your question, the decision to use cefovecin was dictated by the dataset, it was not based on the recommendations of the German Veterinary Society. (lines 121-123, 204-208)

Nevertheless, the study misses to show that cefovecin resistance is comparable to typical 3GC cefotaxime or ceftazidime or that cefovecin resistance is mediated by ESBL/AmpC beta-lactamases. At least the authors could have shown, that ESBL/AmpC-producing E. coli show phenotypic resistance to cefovecin. As there are authors from the Institute of Microbiology as well as from the Veterinary Centre for Resistance Research, they have the opportunities for this (proven by other publications). 

AU: Dear reviewer, you are completely right that cefovecin is not part of the ESBL diagnostic. As described above and now in the methods section, cefovecin was used as 3GC because it was the only 3GC registered for dogs and cats in Germany. Since our analysis was retrospective and the isolates were not available any more, we could not further investigate them. We included some studies that show the comparability to other 3GC [1,2]. 

(lines 121-123, 204-208)

It is essential to investigate these points, as cefovecin isn’t used in other German surveillance

---

## [Decision Letter · Decision Letter 1]

9 Jun 2024

PONE-D-23-35833R1Towards integrated AMR monitoring: Third-generation cephalosporin resistant Escherichia coli in dogs and cats in Germany in 2019-2021PLOS ONE

Dear Dr. Bartel,

Thank you for submitting your manuscript to PLOS ONE. After careful consideration, we feel that it has merit but does not fully meet PLOS ONE’s publication criteria as it currently stands. Therefore, we invite you to submit a revised version of the manuscript that addresses the points raised during the review process. Your manuscript has been returned to the original reviewers and their comments are enclosed for your reference.Please follow their comments and perform all necessary revision.

We look forward to receiving your revised manuscript.

Kind regards,

Yung-Fu Chang

Academic Editor

PLOS ONE

Reviewers' comments:

Reviewer's Responses to Questions

**Comments to the Author**

1. If the authors have adequately addressed your comments raised in a previous round of review and you feel that this manuscript is now acceptable for publication, you may indicate that here to bypass the “Comments to the Author” section, enter your conflict of interest statement in the “Confidential to Editor” section, and submit your "Accept" recommendation.

Reviewer #1: (No Response)

Reviewer #2: All comments have been addressed

Reviewer #3: (No Response)

2. Is the manuscript technically sound, and do the data support the conclusions?

Reviewer #1: Partly

Reviewer #2: Yes

Reviewer #3: Partly

3. Has the statistical analysis been performed appropriately and rigorously? 

Reviewer #1: Yes

Reviewer #2: Yes

Reviewer #3: I Don't Know

4. Have the authors made all data underlying the findings in their manuscript fully available?

Reviewer #1: Yes

Reviewer #2: Yes

Reviewer #3: No

5. Is the manuscript presented in an intelligible fashion and written in standard English?

Reviewer #1: Yes

Reviewer #2: Yes

Reviewer #3: Yes

6. Review Comments to the Author

Reviewer #1: I appreciate the efforts made by the authors to improve the manuscript and I see most of suggested changes/edits by the reviewers are well addressed. However, I cannot recommend acceptance until you remove the direct comparison of human and animal %s. I did not mean you should not talk about the numbers and compare the "trends"; but directly saying %s in animals are higher than dogs/cats does not make a lot of sense to me (data are from a very different database/design/context, etc.) - I say this as a veterinarian and epidemiologist, so you won't need to explain further about how "One Health" or bonds between pets and human work in your response anymore (if you choose to address my comment). Once you remove/resolve this issue and adjust all the texts around this point (if you choose to), I can recommend 'publication'. Otherwise, I'll step down of this review. Also, there is a clear difference between "rate" and "proportion/risk", while this is not a vital point for the acceptance on my end, I suggest you use term "proportion" or simply "%" rather than "resistance RATE" (you could refer to any basic epidemiology textbook on this). Best wishes!

Reviewer #2: Thank you for addressing all of my comments. I would prefer separating cats and dogs and use the Supplementary figure 1 in the article instead Figure 4.

Reviewer #3: Thanks for your revised manuscript.

Please see review report for recommendations.

Data not fully available due to privacy restriction, but provided data is sufficient .

7. PLOS authors have the option to publish the peer review history of their article (what does this mean?). If published, this will include your full peer review and any attached files.

Reviewer #1: No

Reviewer #2: No

Reviewer #3: No

---

## [Author Response · Author response to Decision Letter 1]

17 Jul 2024

Reviewer #1: I appreciate the efforts made by the authors to improve the manuscript and I see most of suggested changes/edits by the reviewers are well addressed. However, I cannot recommend acceptance until you remove the direct comparison of human and animal %s. I did not mean you should not talk about the numbers and compare the "trends"; but directly saying %s in animals are higher than dogs/cats does not make a lot of sense to me (data are from a very different database/design/context, etc.) - I say this as a veterinarian and epidemiologist, so you won't need to explain further about how "One Health" or bonds between pets and human work in your response anymore (if you choose to address my comment). Once you remove/resolve this issue and adjust all the texts around this point (if you choose to), I can recommend 'publication'. Otherwise, I'll step down of this review. Also, there is a clear difference between "rate" and "proportion/risk", while this is not a vital point for the acceptance on my end, I suggest you use term "proportion" or simply "%" rather than "resistance RATE" (you could refer to any basic epidemiology textbook on this). Best wishes!

AU: Thank you very much for your feedback. We have revised the paper and removed all references to direct comparisons. The One Health paragraph has been changed and some additional sources of transmission have been discussed. AMR should be viewed as a One Health problem, which also means that blaming only one of the sectors is wrong. We know that veterinary medicine is often blamed for high AMR rates, and we understand your concerns. The resistant E.coli in cats and dogs cannot be blamed for problems in hospitals. The German AMRPet study proves this but sadly is still in print at Eurosurveillance and will hopefully be published soon. The database/study design in our study is exactly the same as for the human AMR data from the German ARS database, both a just using routine laboratory data. Even the coverage of 1/3 of practices in Germany is the same. The only reason the remove the human data is the concern that a personally biased reader could willfully misrepresent our results. We don’t think that this should guide what we present in our paper, and we are confident that most scientific readers will understand the complexity of this problem. As we write in our paper, there are of course differences between the application of AST in veterinary and human medicine. Furthermore, the difference in 3GC resistance rate between veterinary and human samples in our study is quite small, which is exemplified by the comparatively larger differences in 3GCR rate between countries.

We have also changed "rates" to "proportions" everywhere. We hope the paper now meets your requirements.

Reviewer #2: Thank you for addressing all of my comments. I would prefer separating cats and dogs and use the Supplementary figure 1 in the article instead Figure 4.

Thank you for reviewing the paper again. We have replaced figure 4 with the supplementary figure 1.

Reviewer #3: Thanks for your revised manuscript.

Please see review report for recommendations.

Data not fully available due to privacy restriction, but provided data is sufficient .

AU: Thank you for your recommendations. We have addressed and implemented all the points you requested. The One Health paragraph has been changed and expanded. See below for the individual responses to your recommendations. We hope the paper now meets your expectations.

Abstract:

Method-Section: 

Line 37-38: you can do AST of bacterial strains/isolates but not from a sample, maybe “AST form isolates obtained from samples collected…”?

AU: Thank you for your comment. It has been changed. 

Line 40: the number of E.coli isolated is not relevant for the paper; you provide results for 25.404 E. coli with an MIC, so this should be the number in the abstract, as later in the abstract, the rates given were calculated based on this

AU: Thank you for your feedback. This sentence has been changed. 

Result part: question on the rates, please see in the main text review

Conclusion part: Line 52-55 I would suggest to re-write the sentence e.g. …to their owners and transmission of 3GCR between them is likely as well as acquisition from other (environmental) sources. [Comparison to human data here is too separated, as there can’t be a discussion on this fact here and as you do not know about the bias of pre-treatments, duplicates etc…]

AU: We changed the sentence according to your suggestion. 

Introduction:

Line 68-70: I would shift this to the end of the first paragraph, it is here off-context and fits better to the transmission mentioned above.

AU: Thanks for that comment. We have aggregated the first two paragraphs in the introduction. 

Method:

Line121: “preparation” (small letters)

AU: Has been changed.

Results: 

In- and exclusion criteria for the study should be part of the method section as well as number of isolated E. coli per sample. Further you should exclude all isolates for analysis of all microbials of which data is missing (see figure 3). There are inconsistencies in the number resulting E. coli with accessible MIC; if there are 87 samples with two obviously different E. coli, they count double in the numbers of assessable MIC, so 25.491? The problem here is, that you count samples, but this is the number of E. coli isolates, which makes it difficult to follow also in Table 1. It would be easier to use the number of isolates consequently. Last three lines are confusing, wouldn’t bring them in the table. Just explain in the text, that there are 87 samples with two different E. colis variants isolated and what was done (used both or just one, as your numbers indicate) [all other samples just one E. coli was picked 

AU: Thank you very much. We have changed all the numbers accordingly. The last three lines of the table have been removed and the paragraph above mentions that there were 87 E. coli samples with two isolates. The table now contains only the isolates as requested and all percentages have been adjusted. The text above also mentions which samples were excluded due to lack of growth or lack of MIC.

Figure 2 makes only sense, if the information is connected to something else. Land use, agriculture, population density, WWTP or whatever; as information is not assessable from ARS, you should proof more than one hypothesis. 

We looked at both population density and livestock density in Germany and added the respective data sources to the figure legend. For livestock density there is some visual relationship, but since this is an ecological relationship at best, we decided to not to draw further conclusions.

Line 186-188 (and Fig3): Numbers to compare rates should be only on isolates where you have all MIC information. Different numbers are confusing and are not useful to compare rates. Further you should compare resistance rates of other antimicrobials than cefovecin between 3GCR and on-GCR. It makes no sense to compare rates of 3GRC with a population including the same isolates. (abstract as well)

AU: Thank you for your comment. We choose to analyse all available data, since analysing only isolates with complete MIC information could introduce bias (depending on missingness mechanism). Overall missingness is below 0.5%, which means rates only change by at most +/- 0.5% when only complete cases are used. For missingness rates this low, comparison is therefore not affected. 

Discussion:

Second paragraph (comparability of FOV to other 3GC) should be moved after the first sentence of the first paragraph. Rest of the first paragraph fits better to the third one. 

AU: Thanks. We rearranged the paragraphs. 

In both paragraphs (actually 1 and 3) you just mention other publications and numbers but a real discussion is missing. Are the different reports comparable (interpretation breakpoints vs ECOFF)? Are there information on pre-treatments and so on given? etc. 

AU: Thank you very much for your feedback. To address your comment, we have added a new paragraph and gone into more detail about the comparability of the studies. We have addressed the following points: Isolation, pre-treatments, sample types, antimicrobial susceptibility testing guidelines. We hope this meets your expectations.

Line 206/207: suggest to re-write the sentence; CTX-M and CMY were detected, doesn’t mean that other resistance mechanism (TEM, SHV) wouldn’t mediated cefovecin resistance (Ref 24 detected them) e.g. “Cefovecin resistance was detected in ESBL as well as AmpC-producing E. coli…”

AU: We changed the sentence accordingly to your suggestion. 

In general: you mention OneHealth but only focussed on humans (weak discussion on human ESBL rates, by the way); there is no real discussion on environment, livestock and food (raw meat especially for dogs a definite possible source). Tip: by comparing your data (resistance rates) with literature, be sure to use data from the same isolation procedure; do not compare selective isolation (e.g. ESBL/AmpC monitoring), instead have a look on indicator E. coli monitoring / studies. But, as selective isolation result in higher detection rates, don’t forget to discuss this and the meaning for your data. 

AU: Thank you for your comment. We have tried to better incorporate and consider the various sources of transmission such as raw meat diet, environmental contact, animals and humans in our discussion to reflect the OneHealth aspect. We have excluded the selective isolation study.

Line 218: Paragraph on GermVet here and also later in the discussion; suggest to combine both to avoid redundancies

AU: Thanks for your suggestion. We would like to keep these two paragraphs separate because the first mention of Germ-Vet is about the fact that it is the only monitoring system in Germany that evaluates AMR data from dogs and cats, and we wanted to present the current numbers. In the second mention, Germ-Vet should only serve as an example of active monitoring to illustrate the differences between active and passive monitoring. 

Line240: a) please use the original reference, this is a book chapter, which should have a refence list (same line 258)

 b) although there is a transmission to environment including soil, the sentence suggests that there is a higher contamination rate in soil and therefore dominant suggested source. Please provide appropriate original research reference and discuss further points (see above)

AU: We checked Sobkowich, 2023 (Line 240) and Zogg, 2018 (Line 258) and found no referenced book chapters. Both papers present this information as their own. With regard to the paragraph about the soil, we have changed the text and incorporated other sources of transmission as you suggested.

Line 244-249: Paragraph out of context, could be integrated to second paragraph for comparability of FOV to other 3GC 

AU: Thank you for your comment. We combined this paragraph with the paragraph above, because we made some more hypotheses for possible sources of AMR transmission and found it fits best here. We hope it is to your satisfaction. 

Line 250-255: see above; an in depth discussion is missing, otherwise this part and figure can be deleted;

AU: See answer above regarding Fig.2. 

Line 262/263: Last sentence has to be revised

AU: We changed the last sentence in this paragraph. 

Line 269: re-write e.g. therefore, detailed and current data in AMR rates is needed…

AU: Thank you for the comment. See answer above.

CAZ: in discussion and conclusion: although you introduced CAZ as abbreviation, whole word would be easier to follow for the reader, especially as there are several abbreviations for ceftazidime used between sectors and methods…

AU: AU: Thanks for the tip. We changed both CAZ and FOV to whole words because you are right, it is easier to follow.

---

## [Decision Letter · Decision Letter 2]

31 Jul 2024

Third-generation cephalosporin resistant Escherichia coli in dogs and cats in Germany in 2019-2021

PONE-D-23-35833R2

Dear Dr.Bartel,

We’re pleased to inform you that your manuscript has been judged scientifically suitable for publication and will be formally accepted for publication once it meets all outstanding technical requirements.

Kind regards,

Yung-Fu Chang

Academic Editor

PLOS ONE

Additional Editor Comments (optional):

Reviewers' comments:

Reviewer's Responses to Questions

**Comments to the Author**

1. If the authors have adequately addressed your comments raised in a previous round of review and you feel that this manuscript is now acceptable for publication, you may indicate that here to bypass the “Comments to the Author” section, enter your conflict of interest statement in the “Confidential to Editor” section, and submit your "Accept" recommendation.

Reviewer #1: All comments have been addressed

Reviewer #3: All comments have been addressed

2. Is the manuscript technically sound, and do the data support the conclusions?

Reviewer #1: Yes

Reviewer #3: Partly

3. Has the statistical analysis been performed appropriately and rigorously? 

Reviewer #1: Yes

Reviewer #3: I Don't Know

4. Have the authors made all data underlying the findings in their manuscript fully available?

Reviewer #1: Yes

Reviewer #3: Yes

5. Is the manuscript presented in an intelligible fashion and written in standard English?

Reviewer #1: Yes

Reviewer #3: Yes

6. Review Comments to the Author

Reviewer #1: (No Response)

Reviewer #3: Dear Authors,

thanks for changing the manuscript according to the recommendatiosn.

just afew comments to the latest versions:

Corresponding author differs between Manuscript and PlosONE header pages

Line 66: just remove “and on a smaller scale” as this is the main driver spreading of ESBLs

Line 110 /112: Endo agar – different spellings; for agars other than TSA you do not provide supplier – in-house prepared?

Line 270:” …that there might be a link”; please change; and provide explanation what kind of link – more animals more ESBL or more human population more ESBL? Higher ownership rates in the areas?

7. PLOS authors have the option to publish the peer review history of their article (what does this mean?). If published, this will include your full peer review and any attached files.

Reviewer #1: No

Reviewer #3: No

---

## [Editor Report · Acceptance letter]

16 Aug 2024

PONE-D-23-35833R2 

PLOS ONE

Dear Dr. Bartel, 

I'm pleased to inform you that your manuscript has been deemed suitable for publication in PLOS ONE. Congratulations! Your manuscript is now being handed over to our production team.

Kind regards, 

on behalf of

Dr. Yung-Fu Chang 

Academic Editor

PLOS ONE